# Loss of fixed nitrogen causes net oxygen gain in a warmer future ocean

Andreas Oschlies [1,2], Wolfgang Koeve [1], Angela Landolfi [1] & Paul Kähler[1]

Oceanic anoxic events have been associated with warm climates in Earth history, and there are concerns that current ocean deoxygenation may eventually lead to anoxia. Here we show results of a multi-millennial global-warming simulation that reveal, after a transitory deoxygenation, a marine oxygen inventory 6% higher than preindustrial despite an average 3 °C ocean warming. An interior-ocean oxygen source unaccounted for in previous studies explains two thirds of the oxygen excess reached after a few thousand years. It results from enhanced denitrification replacing part of today's ocean's aerobic respiration in expanding oxygen-deficient regions: The resulting loss of fixed nitrogen is equivalent to an oceanic oxygen gain and depends on an incomplete compensation of denitrification by nitrogen fixation. Elevated total oxygen in a warmer ocean with larger oxygen-deficient regions poses a new challenge for explaining global oceanic anoxic events and calls for an improved understanding of environmental controls on nitrogen fixation.

[1] GEOMAR Helmholtz Centre for Ocean Research Kiel, Düsternbrooker Weg 20, 24105 Kiel, Germany. [2] Kiel University, 24098 Kiel, Germany. Correspondence and requests for materials should be addressed to A.O. (email: aoschlies@geomar.de)

Among the effects of climate change on the ocean, deoxygenation has relatively recently been identified as a potential threat to marine ecosystems and biogeochemical cycles, in addition to warming and acidification[1–4]. There is ample observational evidence for ongoing deoxygenation[5–7], and concerns have been raised that this may eventually lead to widespread anoxia[8], such as inferred for major mass extinctions associated with warm climate excursions in Earth history[9].

Circulation-biogeochemistry models simulate an accelerating decline in the 21st century marine oxygen inventory for all $CO_2$ emission scenarios used in the Fifth Assessment Report of the Intergovernmental Panel on Climate Change[10]. Extending the simulations to timescales of millennia and longer, however, different models generate qualitatively different projections of oceanic oxygen levels in the far future: A box model predicts a continuous 10–20% decline in the oceanic oxygen inventory over the next few thousand years until it recovers to pre-industrial levels along with atmospheric $CO_2$ and global temperatures on timescales of a hundred thousand years[11]. Three-dimensional biogeochemistry-circulation models[12–14] predict a more rapid recovery of meridional overturning and oxygen within several hundred to a few thousand years.

Until now, the oxygen recovery has been attributed to circulation changes, in particular to a flushing of the deep ocean by enhanced deep-water formation in the Southern Ocean[13,14]. Our complete analysis accounting for all physical and biotic oxygen sources and sinks and their interlinkages with the nitrogen cycle reveals, however, that this alone cannot explain the simulated oxygen gain. We show here that most of the net oxygen gain results from a climate-driven substitution of oxygen-consuming respiration by denitrification and an incomplete compensation by nitrogen fixation, associated with a net loss of fixed nitrogen.

## Results

**Solubility effects**. Due to the temperature control of gas solubility, a future warmer ocean will, on average, dissolve less oxygen. Our numerical model employing a business-as-usual emissions scenario with the emission peak in 2100 and a linear decline of emissions to zero in year 2300 (see "Methods"), simulates a rise in global-mean ocean temperatures by 3.1 °C above pre-industrial until year 3380 (Fig. 1a). The associated solubility changes explain an oxygen decline by 27 Pmol (or 9%), as shown by the abiotic oxygen tracer (Fig. 1b). This amounts to about half the simulated total oxygen loss at the time of the $O_2$-inventory minimum (Fig. 1c) and persists until the end of our experiment.

**Changes in overturning**. Consistent with earlier studies[12,14–16], our model simulates an initial reduction in the meridional overturning circulation, which recovers after several centuries and reaches a maximum near year 3080 to eventually level off at an intensity slightly higher than preindustrial (Fig. 2a). Explanations for such a recovery include a shift to enhanced deep-water formation in the southern hemisphere and an associated shoaling of the overturning circulation associated with North Atlantic Deep Water by about 1000 m. This shoaling also explains the increase in globally-averaged ideal age in spite of more vigorous overturning (Fig. 2a). Ideal age increases by a few hundred years at mid-depth in the North Atlantic and Pacific Ocean (Fig. 3c, d). Changes in simulated oxygen in year 8000 relative to year 2000 reveal, on average, elevated oxygen concentrations below about 1000 m (Fig. 3b), located mostly in the Southern Ocean and the Indo-Pacific region, whereas oxygen levels tend to decrease in the Atlantic and the Arctic Ocean (Fig. 3a).

**Changes in biological production**. Net community production (NCP) describes the surplus of primary production over all

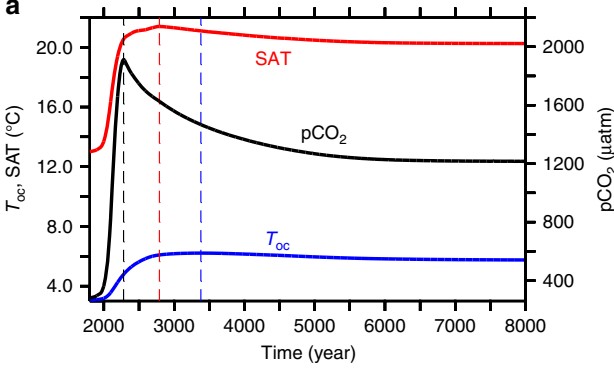

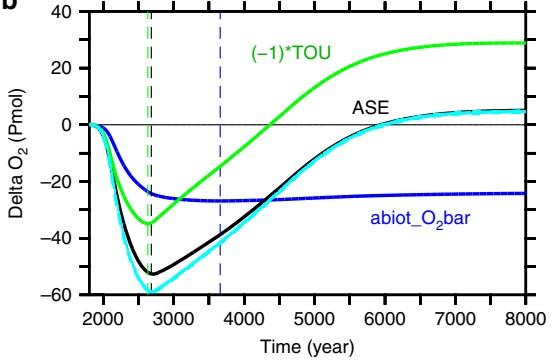

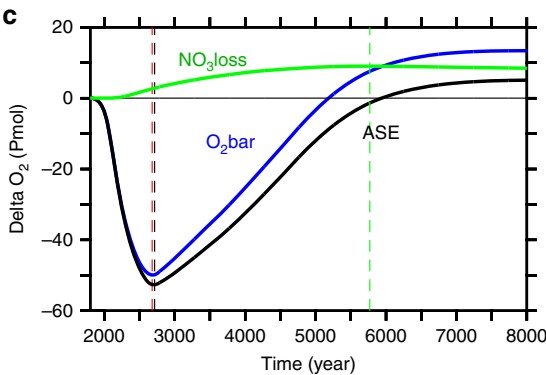

**Fig. 1** Simulated temporal evolution of global indicators. **a** Atmospheric $pCO_2$ (black), global-mean surface air temperature (red) and global mean ocean temperature (blue); **b** anomalies relative to pre-industrial of the global-mean of an abiotic oxygen tracer (blue), total oxygen utilisation (times −1, green), cumulative air-sea oxygen exchange (black; positive into the ocean) and the sum of abiotic tracer and total oxygen utilisation (dashed cyan); and **c** oxygen accumulation via nitrate loss (green), equivalent to the difference between global mean oxygen change (blue) and cumulative air-sea oxygen flux (black). Dashed vertical lines denote the time of maxima/minima of the respective curve of same colour

respiratory processes in the euphotic zone, which, on annual and longer time scales and global space scales corresponds to the organic matter exported from surface water. NCP initially declines by 12% until year 2145, then recovers, and eventually exceeds pre-industrial levels by about 3% (Fig. 2b). The minimum in NCP occurs just when the strength of the meridional overturning circulation is at its minimum (Fig. 2a). Once NCP increases again, the suboxic ($O_2 < 5 \mu M$) volume starts to expand almost threefold within a few centuries (black curve, Fig. 2c). Consequently, pelagic denitrification and, with a time lag of about 25 years, nitrogen fixation both increase and eventually level off at global rates about twice the pre-industrial rates (green and blue

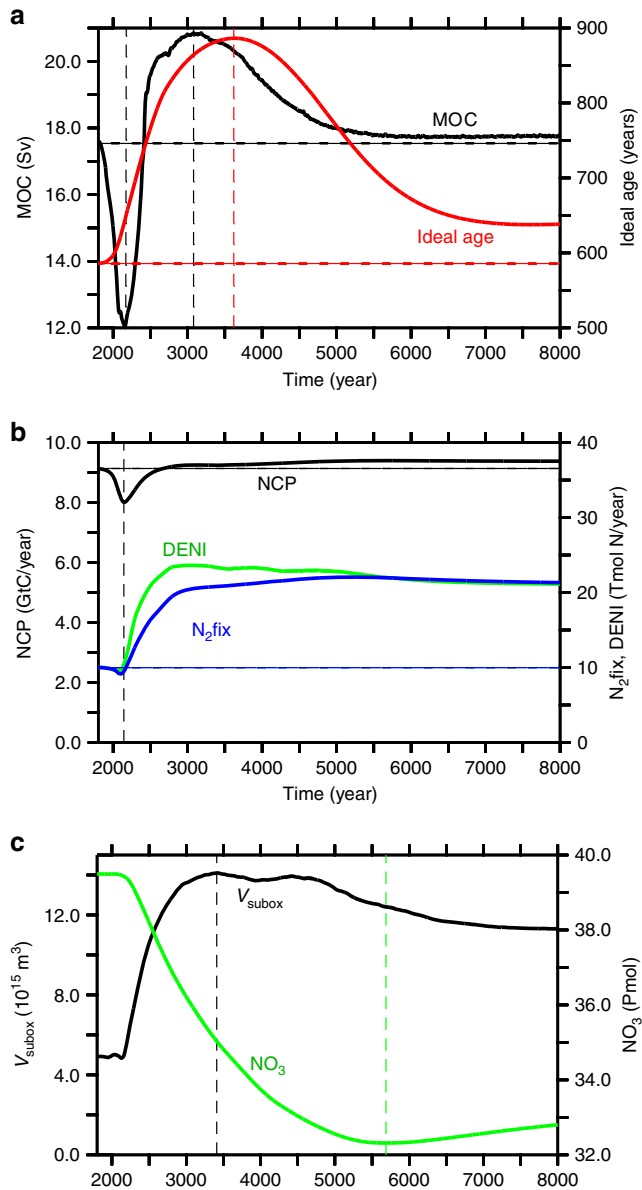

**Fig. 2** Simulated temporal evolution of ocean state indicators. **a** Maximum of the meridional overturning stream function (black) and global average of ideal age (red); **b** global net community production (black), global denitrification (green), global nitrogen fixation (blue); and **c** suboxic volume (black) and globally averaged nitrate concentrations (green)

curves, Fig. 2b). At the end of the simulation, nitrogen fixation accounts for about 20% of the global NCP, compared to about 10% in year 2000.

**Changes in respiratory oxygen consumption**. At first sight, increases in global-mean ocean temperature (Fig. 1a), ideal age (Fig. 2a) and NCP (Fig. 2b) all suggest a declining global ocean oxygen inventory, as also put forward to explain past oceanic anoxic events[17]. The actually modelled oxygen increase (Fig. 1c) can only arise from spatially inhomogeneous changes of temperature, ventilation, respiration, or other processes not yet considered. Indeed, changes in respiratory oxygen consumption via aerobic remineralisation of detritus show a systematic shift of enhanced export and subsequent remineralisation towards high latitudes (Fig. 3g, h) linked to more efficient nutrient utilisation

via improved light conditions due to disappearing summer sea-ice and more stable stratification. Interestingly, particle export and subsequent remineralisation are also enhanced in the upper hundred metres of the subtropical gyres, while there is a general decline of respiration in less well ventilated deeper waters (Fig. 3h). In the model, this results from faster—and hence shallower—remineralisation at higher temperatures, leading to enhanced nutrient recycling in the upper ocean.

**Accumulation of oxygen deficits**. Although total respiration shows a slight increase, the oceanic storage of the respiratory oxygen deficit, measured here as true oxygen utilization (TOU[18]), is decreased considerably in the simulated future ocean. For the first few hundred years of our simulation, TOU increases ((−1) *TOU decreases, green line in Fig. 1b). This is because of the longer residence time of upwelled waters in the more stratified surface waters allowing for a more complete biotic utilisation of surface nutrients, and elevated oxygen consumption during subsequent remineralisation in the ocean interior. TOU decreases ((−1)*TOU increases) after year 2630 and eventually the global TOU inventory becomes lower than the pre-industrial one. The total reduction in TOU amounts to an oxygen gain of 29 Pmol by the end of the simulation. This more than compensates the warming-driven oxygen loss from solubility changes of 24 Pmol, yielding a net oxygen surplus of 4.6 Pmol. This extra oxygen must enter the ocean via air-sea exchange. Indeed, the cumulative air-sea oxygen flux into the ocean by the end of the simulation (5.1 Pmol, Fig. 1b, black line) very closely matches the combined oxygen loss via warming-induced solubility change and the oxygen gain by the decrease of TOU (Fig. 1b, dashed cyan line).

**Effects of nitrogen imbalances**. The oxygen gained via air-sea gas exchange explains, however, only about one third of the total increase in the oceanic oxygen inventory (13.4 Pmol, i.e. 6% of today's ocean $O_2$ inventory Fig. 1c, blue line). Almost two thirds of it must therefore be from an oxygen source located in the ocean interior. Interior-ocean oxygen sources are photosynthesis, specifically, the net balance of oxygenic photosynthesis minus respiration, and processes reducing the oxidation state of nitrogen and other compounds. For everything else unchanged, a switch from aerobic to anaerobic respiration will cause a left-over of, and hence increase in, dissolved oxygen. Total marine oxygen may thus be saved by the utilization of an oxidant other than oxygen, like in denitrification, or it may be spent by oxidising an external source of reduced nitrogen, like in $N_2$-fixation and subsequent nitrification of the newly added N. In our simulation the strong increase in pelagic denitrification together with the time-lagged increase in $N_2$-fixation cause a net loss of the ocean's fixed nitrogen (Fig. 2c). Any net conversion, and loss, of nitrate to $N_2$ via denitrification corresponds to a net gain of oxygen (Fig. 1c) because no oxygen is consumed in it. Anaerobic remineralization via denitrification of organic matter containing one mole of organic nitrogen avoids the consumption of 10.6 moles of oxygen while removing 7.48 moles of $NO_3$ (ref. [19]). Thus, for each mole of nitrate lost, about 1.4 moles of oxygen are gained. Denitrification is hence an implicit source of oxygen to the ocean. The reverse applies to $N_2$-fixation which is hence an oxygen sink for the ocean. What matters globally is the cumulative balance of denitrification and $N_2$-fixation, that is the change in the global nitrate inventory. The total simulated nitrate loss of 6.2 Pmol $NO_3$, i.e. about 17% of the present-day ocean's nitrate inventory (green curve Fig. 2c), corresponds to an oxygen gain of 8.8 Pmol $O_2$ (Fig. 1c), which is about 4% of the present ocean oxygen inventory.

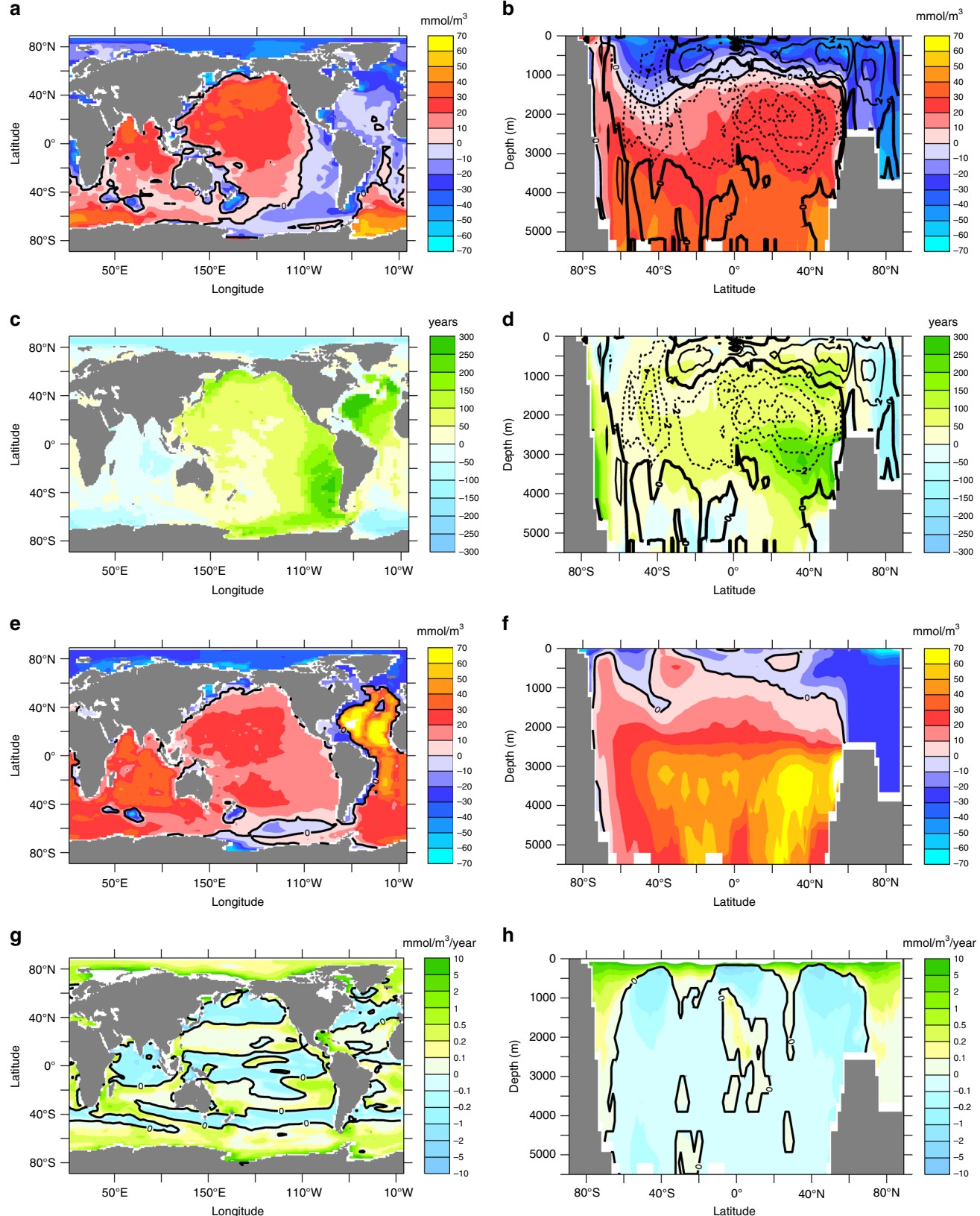

**Fig. 3** Simulated ocean changes (year 8000 minus year 2000). **a**, **b** Oxygen in mmol $O_2$ m$^{-3}$; **c**, **d** ideal age in years; **e**, **f** total oxygen utilisation in mmol $O_2$ m$^{-3}$; and **g**, **h** remineralisation below 125 m in mmol $O_2$ m$^{-3}$ year$^{-1}$. The left hand column (panels **a**, **c**, **e**, **g**) shows vertically averaged changes, the right hand column shows zonally averaged changes. Contour lines in panels **b** and **d** refer to the zonally integrated meridional overturning stream function with transport in units Sv ($10^6$ m$^3$ s$^{-1}$)

Sulfate reduction to hydrogen sulfide, not considered in the model, would save still more oxygen in the water column. However, unless outgassing to the atmosphere, hydrogen sulfide would most likely quickly be re-oxidised in the oxic waters surrounding the respective anoxic region, cancelling the initial oxygen gain. Not resolving the marine sulfur cycle, we cannot quantify to what extent the reaction of $H_2S$ with $NO_3$, the formation of elemental sulfur, or the burial of FeS might modify this picture. Other processes that may affect oxygen, but are not explicitly accounted for in this model configuration, are burial and denitrification in the sediments as well as the release of sedimentary phosphate and iron under expanding oxygen-deficient regions[20].

**Nitrogen fixation feedback**. The enhanced loss of fixed nitrogen via denitrification can only lead to a sustained net decline in the ocean's inventory of fixed nitrogen, because nitrogen fixation only partly compensates the nitrogen loss. This, apparently, is contrary to the geochemical view that, on long timescales, nitrogen fixation feeds back on nitrogen losses and 'nitrate gets topped up when scarce relative to phosphate' (ref. [21]). In our simulation, nitrogen fixation is performed by diazotrophs with a maximum growth rate less than half of that of ordinary phytoplankton and zero growth at temperatures lower than 15 °C, but not limited by fixed nitrogen[12]. It starts to increase within 25 years after the onset of the rise in denitrification, but does so at a slower rate catching up with annual rates of nitrogen loss only well after year 5000 (Fig. 2b). The initial delay is consistent with the time it takes for water to upwell from the oxygen minimum zones (with denitrification) to the surface[22]. Simulated global nitrogen fixation more than doubles within a few hundred years (Fig. 2b). It

expands its range with the poleward migration of the 15 °C isotherm (Fig. 4a). Areas of enhanced denitrification, however, remain located in the tropical oceans (Fig. 4b). As shown in Figs. 4c, d, lowest values of $N^* = NO_3 - 16\ PO_4$ (ref. [23,24]) that measures the nitrate excess relative to the stoichiometric equivalent of phosphate, are simulated in regions of ongoing denitrification in the tropical oceans, whereas positive $N^*$ values are found in the surface waters of the tropical and subtropical oceans, roughly coinciding with areas of nitrogen fixation (Fig. 4a). Surface waters at high latitudes, particularly in the Southern Ocean, and almost all of the deep ocean waters, however, display negative $N^*$ values. This illustrates that part of the denitrification signature is upwelled in the Southern Ocean, where high nutrient levels and low surface temperatures and light levels do not present favourable conditions for nitrogen fixation. Deep-water formation south of the Southern Ocean biogeochemical divide[25] ensures that any upwelled low-$N^*$ waters can be transported into the deep ocean without supporting nitrogen fixation. It thereby acts as a loophole through which denitrification-induced nitrogen deficits can escape the surface-ocean nitrogen-fixation feedback and preserve significant oceanic nitrate deficits in the ocean interior.

**Decoupling of biological production and nutrient inventory**. An interesting finding is that global NCP is essentially unchanged and even shows a slight increase (Fig. 2b) despite a 17% decline in the ocean's nitrate inventory (Fig. 2c). This decoupling of global NCP and nutrient inventory can be explained by the spatial pattern of the accumulated nitrate deficit, which very closely corresponds to the pattern of $N^*$ (Fig. 4c, d): Nitrate concentrations are lowered predominantly in the deep ocean and in

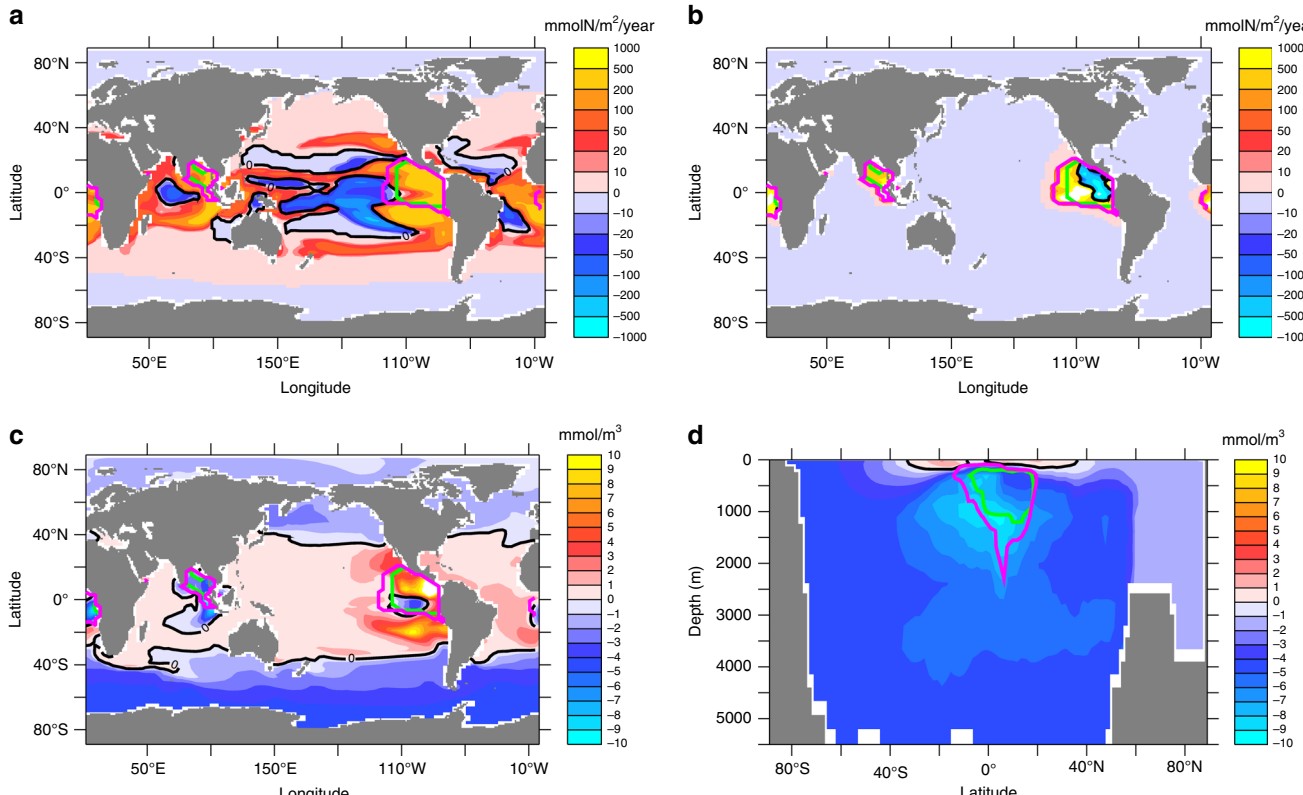

**Fig. 4** Simulated nitrogen-cycle changes (year 8000 minus year 2000). **a** Annual mean vertically integrated nitrogen fixation, **b** annual mean vertically integrated denitrification, **c** surface concentrations in $N^* = NO_3 - 16\ PO_4$, and **d** zonally averaged $N^*$. Green contours refer to the maximum extent of suboxic waters ($O_2 < 5\ \mu M$) in year 2000, magenta contours to the maximum extent of suboxic waters in year 8000

high-latitude surface waters, whereas nitrate concentrations do not change significantly or even increase in the tropical and subtropical surface layers where phytoplankton growth is typically limited by nitrogen[26].

## Discussion

Our modelling study of global-warming effects over millennia suggests expanding ocean anoxia (and the associated switch from aerobic respiration to denitrification) to eventually result in a counterintuitive net increase of marine oxygen levels. For each mole nitrate consumed via denitrification, about 1.4 moles of $O_2$ are saved. The same amount of oxygen is consumed when organic nitrogen stemming from nitrogen fixation is oxidised to nitrate. For a net oxygen gain it is essential that the commonly assumed feedback between nitrogen fixation and denitrification operates only with a temporal lag. The imbalance between nitrogen fixation and denitrification lasts for about 3000 years in our model simulation (Fig. 2b). In our model this is by the entrainment of nitrate-deficient waters into deep-water formation regions, particularly in the Southern Ocean. Via this loophole, nitrate deficits can escape the topping-up action of nitrogen fixation and instead remain stored in the deep ocean, effecting a net oxygen gain. A major uncertainty (and on the agenda for future research) are the environmental controls of nitrogen fixation. Sensitivity experiments performed with a mechanistically different model of nitrogen fixation that includes the ability to access dissolved organic phosphorus and also allows diazotrophs to grow at low temperatures[27], yield similar results regarding the temporal lag of the nitrogen fixation, but with only about half the simulated total loss of nitrate and associated gain of oxygen. If current assumptions about the environmental controls of nitrogen fixation are correct and properties of deep ocean waters can be set relatively unaffected by nitrogen fixation, the process modelled here leading to a net oceanic oxygen gain for a long-term global warming scenario may well have operated also during the development of whole-ocean anoxic events, e.g. in the Cretaceous. It seems to be more difficult to develop marine anoxia than thought until now.

## Methods

**Model configuration**. The model used is the University of Victoria (UVic) Earth System Climate Model[28] in the configuration described by ref. [12]. The ocean component is a fully three-dimensional primitive-equation model with nineteen levels in the vertical ranging from 50 m near the surface to 500 m in the deep. It contains a simple marine ecosystem model including the two major nutrients nitrate and phosphate and two phytoplankton classes, nitrogen fixers and other phytoplankton, the former being limited by phosphate only. As a caveat we note that the micronutrient iron is not explicitly included in the model, which nevertheless achieves a reasonable fit to observed biogeochemical tracer distributions for the tuned biological parameters and mixing parameterizations[12,29]. For the molar stoichiometry of C:N:P:$-O_2 = 112:16:1:169.6$ assumed in the model[19], organic matter is degraded by aerobic remineralisation ($-O_2:PO_4 = 169.6$) as long as sufficient dissolved oxygen is available. In regions where oxygen concentrations are below a threshold of 5 mmol $O_2$ m$^{-3}$, nitrate is used as electron acceptor (denitrification, $-NO_3:PO_4 = 119.68$). No other electron acceptors are simulated and remineralisation stops whenever nitrate runs out, which happens only occasionally at very few grid points. A suite of idealised tracers is added comprising ideal age[30], an abiotic oxygen tracer affected only by air-sea gas exchange and surface-water solubility[31], and a tracer of preformed $PO_4$ that is, at every model time step, set identical to the model's $PO_4$ tracer in the surface layer, but otherwise a passive tracer without sinks or sources[32]. TOU is computed as the stoichiometric oxygen equivalent of regenerated $PO_4$, i.e. total $PO_4$ minus preformed $PO_4$.

The ocean component is coupled to a single-level energy-moisture balance model of the atmosphere and a dynamic-thermodynamic sea ice component and a terrestrial vegetation and carbon-cycle component[28]. All model components use a common horizontal resolution of 1.8° latitude times 3.6° longitude. The current model version does not consider any fluxes across the water-sediment interface and also does not account for fluxes related to weathering on land. Oceanic phosphorus is thus strictly conserved. Because the atmosphere contains about a hundred times as much oxygen as the ocean, any feedback of marine oxygen changes on atmospheric oxygen is neglected as in earlier studies (e.g. ref. [11]).

**Global warming scenario**. After a spin-up of more than 10,000 years under pre-industrial atmospheric and astronomical boundary conditions, the model is run under historical conditions from year 1850 to 2000 using fossil-fuel and land-use carbon emissions as well as solar, volcanic and anthropogenic aerosol forcings. From year 2000 to 2100, the model is forced by $CO_2$ emissions following the Special Report on Emissions A2 non-intervention scenario[33] reaching maximum emissions of about 29 PgC/year in the year 2100. Thereafter, $CO_2$ emissions decrease linearly to zero in year 2300. In this simulation, 2200 GtC are emitted until year 2100 and a total of 5100 GtC until year 2300. Model simulations are continued with zero $CO_2$ emissions after year 2300 until year 8000.

## Data availability
Model results generated as part of this study are available at https://data.geomar.de/thredds/catalog/open_access/oschlies_et_al_2019_nc/catalog.html

## Code availability
All code used to generate the results of this study is available at https://data.geomar.de/thredds/catalog/open_access/oschlies_et_al_2019_nc/catalog.html

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

## Acknowledgements
This work was supported by the German Research Foundation (DFG) as part of the research project SFB 754 'Climate-Biogeochemistry Interactions in the Tropical Ocean'.

## Author contributions
All authors discussed the results and wrote the manuscript. A.O. designed the study, performed the simulations and led the analysis and writing. W.K., A.L. and P.K. contributed to analysis, interpretation of results and writing.

## Additional information

**Competing interests:** The authors declare no competing interests.

