## [Peer Review File · Nature Communications]

Reviewers' comments:

Reviewer #1 (Remarks to the Author):

The paper presents an interesting analysis of the future evolution of dissolved oxygen (DO) under global warming. The study identifies the important role the nitrogen cycle (N-fixation and denitrification) can play in evolution of DO. The study has important implications for both past changes in oceanic DO (like the last glacial maximum) and the long term evolution of DO with global warming. The paper is novel and well written and it presents an important result and I recommend publication with minor revisions.

Few minor comments to consider with the revision

Abstract - very important to emphasise that both the volume of suboxic water and the denitrification increase with the later working to increase global inventory of oxygen through its impact on organic matter remineralisation

lines 33-47 - important to include the nitrogen cycle in the model to capture the interactions between DO changes and N-cycling.

Question- how is carbon cycling altered?

Model Setup- does the model consider benthic processes? discuss how benthic processes could alter the link between DO and the nitrogen cycle.

line 189 - Link between N-fixation and Denitrification - important to mention this imbalance is only temporary (how long does it take to equilibrate? 8000years). The adjustment time may have important implications for how we interpret paleo observations of changes in the volume of suboxic water and the impact of N-fixation and Denitrification.

Is there a relationship between volume of suboxic water and De-Nitrification, or N-fixation? plot them. How do they compare spatially?

Figure 4. - maximum extent of suboxic water - year 8000? state what year what does it look like at year 2000? Could add this to the figures.

Reviewer #2 (Remarks to the Author):

This manuscript describes model-predicted changes in ocean circulation and biogeochemistry on millennial time scales (2000 – 8000 yr) that result from fossil fuel CO₂ production levels projected in IPCC reports. The focus is on ocean oxygen, and fixed nitrogen content. The model output describes a situation in which oxygen levels decrease in the next few hundred years, as expected, but then recover and actually exceed today's levels after about the year 5000. The reason for the model projections are the increase in denitrification and decrease in NO₃⁻ inventory during low oxygen levels. In the situation where carbon flux to the deep ocean is constant and there is no carbon burial, an increase in the fraction of NCP respired by denitrification actually causes a small increase in the oxygen concentration using stoichiometries of the oxygen and denitrification respiration reactions. I

believe this is correct and has not been pointed out before. It could be an important contribution to understanding of the long-term ocean response to global warming.

The entire result of this model study rests on the lag in nitrogen fixation following the increase in denitrification. This is what causes the decrease in NO_3^- inventory and hence the stoichiometric increase in O_2 inventory. This is described by the authors. However, there is another issue here that does not receive much attention. That is that the NCP (or the flux of organic C to the deep ocean) changes during the initial response to decreased circulation, but after that it remains almost constant. Normally, geochemists think that NCP is a function of the nitrate content of the ocean, because NO_3^- is the long-term limiting nutrient, and upwelling to the surface ocean limits NCP. What one might expect is that when NO_3^- concentrations are lowered, NCP would decrease which would cause a negative feedback that would bring inventories of NO_3^- and O_2 back toward the original levels. In this model run, NCP and NO_3^- inventory are uncoupled. I suppose this is because enhanced nitrogen fixation supplies the N missing from NO_3^- upwelling. But, is this really the way the ocean works?? Do plankton not have a preference for NO_3^- over N_2 as a nitrogen source. In this model run the source of N supporting NCP is apparently almost constant after the first 1000 years.

The authors explain the importance of the nitrogen fixation response to denitrification for the result they present, but do not mention the uncertainty of the competition of different plankton for N from NO_3^- and N_2 . I suspect this is caused by the way the ecosystem is parameterized in the model. I believe the reader is entitled to know how sensitive the results are to these uncertainties. How unique is this result? If you change the nitrogen fixation response within levels believed to be possible how much does it change the result of O_2 inventory after the year 5000? What would happen if NCP and NO_3^- inventory ARE related? I think for these unique and insightful observations to become part of how we perceive future ocean oxygen content to evolve the authors should have to show the sensitivity of their result to the uncertainties of the nitrogen cycle.

Rebuttal letter

We thank the reviewers for their careful reading and their constructive comments and suggestions which, we think, helped to improve the paper. In the following, reviewer comments are marked in *italics*, our responses in roman font.

Reviewer #1 (Remarks to the Author):

The paper presents an interesting analysis of the future evolution of dissolved oxygen (DO) under global warming. The study identifies the important role the nitrogen cycle (N-fixation and de-nitrification) can play in evolution of DO. The study has important implications for both past changes in oceanic DO (like the last glacial maximum) and the long term evolution of DO with global warming. The paper is novel and well written and it presents an important result and I recommend publication with minor revisions.

Few minor comments to consider with the revision

Abstract - very important to emphasise that both the volume of suboxic water and the denitrification increase with the later working to increase global inventory of oxygen through its impact on organic matter remineralisation

We have now added: “It results from **enhanced** denitrification replacing part of today’s ocean’s aerobic respiration in **expanding** oxygen-deficient regions” (line 19/20).

lines 33-47 - important to include the nitrogen cycle in the model to capture the interactions between DO changes and N-cycling.

Regarding the nitrogen cycle, we have included (line 44): “Our complete analysis accounting for all physical and biotic oxygen sources and sinks **and their interlinkages with the nitrogen cycle** reveals...”.

Question- how is carbon cycling altered?

Carbon cycling is affected via Redfield stoichiometry of organic matter (C:N:P=112:16:1) with consistent treatment of anaerobic processes following Paulmier et al. (2009). While denitrification does not consume oxygen, it still produces the same amount of CO₂ as aerobic respiration of organic matter. Switching from aerobic to anaerobic respiration has therefore no direct impact on the carbon cycle. However, there are indirect impacts, in particular:

- (i) anaerobic remineralisation via denitrification is slower (at least in our model) than aerobic remineralisation (Fig.1 of Paulmier et al., 2009), leading to a deeper export of organic carbon.
- (ii) a reduced ocean inventory of fixed N will tend to increase N limitation, which can affect patterns of primary production and subsequent export production (despite the global uncoupling of NCP and N inventory).

- (iii) production of alkalinity via denitrification reduces the local partial pressure of CO₂ in sea water

All effects are consistently accounted for in the model, but are not expected to become a major effect on oceanic carbon uptake in a high-CO₂-emission world simulated here, where changes in the oceanic carbon uptake are dominated by the solubility pump. We therefore think that adding a discussion on carbon-cycle effects would distract from the main story of at first sight unexpected oxygen changes.

Model Setup- does the model consider benthic processes? discuss how benthic processes could alter the link between DO and the nitrogen cycle.

Benthic processes are not resolved by the model. All organic matter sinking to the sea floor will remain there until remineralised with the temperature-dependent and oxygen-dependent remineralisation rate of particulate organic matter in the water column. Burial and/or benthic denitrification of organic carbon would consist in another net oxygen source that is neglected in this study. Current estimates amount to a total contribution of 0.002 Pmol O₂/yr (Wallmann, 2010), and changes in benthic process will likely only amount to a fraction of this. Even the total amount is smaller than the oxygen source that results from enhanced water-column denitrification in expanding oxygen-deficient waters in the model simulation (about 0.005 Pmol O₂/yr between year 2500 and 4500, Fig.1c). Burial would decouple oxygen source and nitrogen cycle, benthic denitrification would, via exchange with bottom waters, couple oxygen source and nitrogen sink in essentially the same way as pelagic denitrification.

Changes in the redox state of bottom waters in expanding oxygen-deficient regions could lead to enhanced release of phosphorus and iron. This is another feedback loop not accounted for here that might, via the supply of additional nutrients, lead to enhanced oxygen consumption (Niemeyer et al., 2017; Kemena et al., 2018).

We have added (lines 152-156): **Other processes that may affect oxygen, but are not explicitly accounted for in this model configuration, are burial and denitrification in the sediments as well as release of sedimentary phosphate and iron under expanding oxygen-deficient regions.**

line 189 - Link between N-fixation and Denitrification - important to mention this imbalance is only temporary (how long does it take to equilibrate? 8000years). The adjustment time may have important implications for how we interpret paleo observations of changes in the volume of suboxic water and the impact of N-fixation and Denitrification.

Good point. The imbalance is maintained only for about 3000 years (Fig. 2b) until N₂ fixation reaches essentially the same rates as denitrification. This is already mentioned in a previous paragraph in the manuscript (line 167-169). To emphasise this important point further, we have added (lines 204-206): “For a net oxygen gain it is essential that the commonly assumed feedback between nitrogen fixation and denitrification is ~~not operating perfectly~~ **operates only with a temporal lag. The imbalance between nitrogen fixation and denitrification lasts for about 3000 years in our model simulation (Fig. 2b).**”

Results of a sensitivity experiment with a different parameterisation of nitrogen fixation (see response to reviewer 2 and Fig. R3 below) show essentially the same time lag.

Is there a relationship between volume of suboxic water and De-Nitrification, or N-fixation? plot them. How do they compare spatially?

Temporally, the expansion of suboxic water volume and the increase in denitrification and nitrogen fixation are well correlated, with denitrification almost always exceeding the rate of nitrogen fixation during the transient phase (see Figure R1 below). This information is already contained in Fig.2c and 2b, so that we think that no additional figure is needed in the main manuscript. Spatial patterns are shown in Fig.4a for the increase in nitrogen fixation, in Fig.4b for the change in denitrification. In the revised version of Fig.4 we have included the extent of suboxic waters for both year 2000 (green contours) and year 8000 (magenta contours) and updated the figure caption accordingly.

Fig.R1: Scatterplot of denitrification (green) and nitrogen fixation (blue) against the temporal evolution of the suboxic volume (x-axis). Arrows denote the direction of time. From the lower left to the upper right it takes about 500 years, with denitrification continuously exceeding nitrogen fixation.

Figure 4. - maximum extent of suboxic water - year 8000? state what year what does it look like at year 2000? Could add this to the figures.

Thanks for pointing this out. The extent of suboxic waters was shown for year 8000 (green contour in original figure). We have now added the extent of suboxic waters in year 2000 of

the model simulation (**green contours in the revised Fig.4**) in addition to the extent in year 8000 (**magenta contours in the revised Fig.4**).

Reviewer #2 (Remarks to the Author):

This manuscript describes model-predicted changes in ocean circulation and biogeochemistry on millennial time scales (2000 – 8000 yr) that result from fossil fuel CO₂ production levels projected in IPCC reports. The focus is on ocean oxygen, and fixed nitrogen content. The model output describes a situation in which oxygen levels decrease in the next few hundred years, as expected, but then recover and actually exceed today's levels after about the year 5000. The reason for the model projections are the increase in denitrification and decrease in NO₃⁻ inventory during low oxygen levels. In the situation where carbon flux to the deep ocean is constant and there is no carbon burial, an increase in the fraction of NCP respired by denitrification actually causes a small increase in the oxygen concentration using stoichiometries of the oxygen and denitrification respiration reactions. I believe this is correct and has not been pointed out before. It could be an important contribution to understanding of the long-term ocean response to global warming.

We thank the reviewer for this positive assessment. In the revised text we emphasise now more explicitly (line 128-130) **“For everything else unchanged, a switch from aerobic to anaerobic remineralisation will cause a left-over and hence increase in dissolved oxygen.”** In fact, NCP and thus the carbon flux leaving the upper ocean and being remineralised at depth are not exactly constant, but respond to the warming-driven changes in nutrient and light supply and to changes in temperature and oxygen concentrations (via the temperature- and oxygen-dependent respiration rates, Fig.1 of Paulmier et al., 2009).

The entire result of this model study rests on the lag in nitrogen fixation following the increase in denitrification. This is what causes the decrease in NO₃⁻ inventory and hence the stoichiometric increase in O₂ inventory. This is described by the authors. However, there is another issue here that does not receive much attention. That is that the NCP (or the flux of organic C to the deep ocean) changes during the initial response to decreased circulation, but after that it remains almost constant. Normally, geochemists think that NCP is a function of the nitrate content of the ocean, because NO₃⁻ is the long-term limiting nutrient, and upwelling to the surface ocean limits NCP. What one might expect is that when NO₃⁻ concentrations are lowered, NCP would decrease which would cause a negative feedback that would bring inventories of NO₃⁻ and O₂ back toward the original levels.

We agree with the reviewer that the relative insensitivity of NCP to the substantial loss of fixed nitrogen is at first sight surprising. Nitrate concentrations are, however, not lowered everywhere. In fact, the decline in nitrate (as in N^{*}, Fig.4) is mostly seen in the ocean interior and in high-latitude surface waters, whereas nitrate concentrations do not change significantly or even increase in the tropical and subtropical surface layers (Fig.R2) where

phytoplankton growth is limited by N (Moore et al., 2013). While nitrogen fixation appears to fully compensate denitrification-driven losses of fixed nitrogen in the mid- and low-latitude surface ocean, nitrate-deficient waters escape into the deep ocean unaffected by the action of nitrogen fixation. This escape occurs mainly in the Southern Ocean and leads to storage of nitrate deficient waters in the deep ocean interior, thereby decoupling NCP from the global nutrient inventory.

Fig.R2: Differences in nitrate (a,c) and phosphate (b,d) concentrations between model year 8000 minus year 2000 at the ocean surface (a,b) and zonally averaged (c,d).

In the revised manuscript we have now added a brief paragraph of this interesting aspect (lines 188-196).

In this model run, NCP and NO₃⁻ inventory are uncoupled. I suppose this is because enhanced nitrogen fixation supplies the N missing from NO₃⁻ upwelling. But, is this really the way the ocean works?? Do plankton not have a preference for NO₃⁻ over N₂ as a nitrogen source. In this model run the source of N supporting NCP is apparently almost constant after the first 1000 years.

In agreement with the reviewer's (and our) notion, phytoplankton has a preference for NO₃ over N₂ – in our model this is ensured by the faster growth rate of ordinary phytoplankton (exclusively using NO₃ as nitrogen source) compared to diazotrophs, and also by diazotrophs taking up NO₃ preferentially over N₂ (line 164-167). The upper ocean has essentially adjusted after about 1000 model years, after which only the deep (unproductive) ocean fills up with NO₃-deficient waters entering predominantly via deep water formation in the Southern Ocean.

The authors explain the importance of the nitrogen fixation response to denitrification for the result they present, but do not mention the uncertainty of the competition of different plankton for N from NO₃⁻ and N₂. I suspect this is caused by the way the ecosystem is parameterized in the model. I believe the reader is entitled to know how sensitive the results

are to these uncertainties. How unique is this result? If you change the nitrogen fixation response within levels believed to be possible how much does it change the result of O2 inventory after the year 5000?

We agree with the reviewer that this is an important issue. Until now there is only limited mechanistic understanding regarding the environmental controls of nitrogen fixation in the global ocean. Essentially all conceptual and numerical models assume that diazotrophs have an ecological advantage in warm NO₃-deficient waters (relative to PO₄). One exception is the model of Landolfi et al. (2015) that also assumes that diazotrophs have a major advantage in PO₄ depleted waters by being able to invest into N-rich exoenzymes to access DOP. In that model diazotrophs also have a relaxed temperature constraint and are able to grow below 15°C. Landolfi et al. (2015) showed that this hypothesis changed the simulated distribution of nitrogen fixation in the global ocean, with a considerably larger proportion of global N₂ fixation occurring in the N-rich North Atlantic compared to other models and in better agreement with direct measurements (Landolfi et al., 2018). While we do not know whether either of the traditional or the Landolfi et al. (2015) assumptions about environmental controls of nitrogen fixation are correct, both approaches are mechanistically different. To test the robustness of our conclusions to uncertainties in the model parameterisation of nitrogen fixation, we have repeated our simulations with the Landolfi et al. (2015) parameterisation. Encouragingly, the results are very similar (Fig. R3): The increase in denitrification leads that of nitrogen fixation, and nitrogen fixation is lower than denitrification by a few Tmol/yr for more than 1000 years. The cumulative loss of fixed nitrogen is, however, only about half of the loss simulated in the present model. The results nevertheless provide confidence to the general conclusion that the escape of nitrate-deficient waters into the deep ocean via deep-water formation in the Southern Ocean is a robust feature relatively insensitive to assumptions on the environmental controls of marine nitrogen fixation. Light limitation (in addition to iron limitation) in the Southern Ocean provides a physical constraint on the ability of diazotrophs to top up the nitrate deficits generated by enhanced denitrification.

This is now included in the ‘uncertainties and implications’ section in the revised manuscript (line 212-220):

Sensitivity experiments performed with a mechanistically different model of nitrogen fixation that includes the ability to access dissolved organic phosphorus and also allows diazotrophs to grow at low temperatures²⁷, yield similar results regarding the temporal lag of the nitrogen fixation, but with only about half the simulated total loss of nitrate and associated gain of oxygen. If current assumptions about the environmental controls of nitrogen fixation are correct **and properties of deep ocean waters can be set relatively unaffected by nitrogen fixation**, the process modelled here leading to a net oceanic oxygen gain for a long-term global warming scenario may well have operated also during the development of whole-ocean anoxic events, e.g. in the Cretaceous.

Fig.R3: Temporal evolution of denitrification (green) and nitrogen fixation (blue) in a sensitivity experiment using the COST parameterisation of marine nitrogen fixation of Landolfi et al. (2015).

What would happen if NCP and NO₃- inventory ARE related? I think for these unique and insightful observations to become part of how we perceive future ocean oxygen content to evolve the authors should have to show the sensitivity of their result to the uncertainties of the nitrogen cycle.

As shown above, in the three-dimensional ocean, NCP and NO₃-inventory may be decoupled by variations in the vertical structure of the NO₃ distribution. A major uncertainty in the marine nitrogen cycle is the response of nitrogen fixation to changes in the environment. In the absence of a robust mechanistic understanding, this uncertainty has been addressed by re-running the model with a conceptually independent parameterisation of nitrogen fixation. The results of this sensitivity experiment indicate that our results are relatively insensitive to the parameterisation of nitrogen fixation. This is now discussed in the revised manuscript and we believe that this considerably improves the confidence into the robustness of our results (see above and lines 210-220 of the revised manuscript).

References

Kemena, T. P. , Oschlies, A., Koeve, W. , Wallmann, K. J. G. , Landolfi, A. and Dale, A. W. (2018). Ocean phosphorus inventory and ocean deoxygenation: Large uncertainties in future projections on millennial timescales. *Earth System Dynamics Discussions*, 1-28. DOI 10.5194/esd-2018-58.

Landolfi, A., W. Koeve, H. Dietze, P. Kähler, and A. Oschlies (2015). A new perspective on environmental controls of marine nitrogen fixation. *Geophysical Research Letters*, 42, 4482–4489.

Landolfi, A., P. Kähler, W. Koeve, and A. Oschlies. (2018). Global marine N₂ fixation estimates: From observations to models. *Frontiers in Microbiology*, 9:2112.

Niemeyer, D., T. P. Kemena, K. J. Meissner, and A. Oschlies (2017). A model study of warming-induced phosphorus–oxygen feedbacks in open-ocean oxygen minimum zones on millennial timescales. *Earth System Dynamics*, 8, 357–367.

Moore, C. M., M. M. Mills, K. R. Arrigo, I. Berman-Frank, L. Bopp, P. W. Boyd, E. D. Galbraith, R. J. Geider, C. Guieu, S. L. Jaccard, T. D. Jickells, J. La Roche, T. M. Lenton, N. M. Mahowald, E. Maranon, I. Marinov, J. K. Moore, T. Nakatsuka, A. Oschlies, M. A. Saito, T. F. Thingstad, A. Tsuda, and O. Ulloa (2013). Processes and patterns of oceanic nutrient limitation. *Nature Geosci*, 6, 701–710, 2013.

Paulmier, A., I. Kriest, and A. Oschlies (2009). Stoichiometries of remineralisation and denitrification in global biogeochemical ocean models. *Biogeosciences*, 6, 923–935.

Wallmann, K. (2010). Phosphorus imbalance in the global ocean? *Global Biogeochemical Cycles*, 24, GB4030.

REVIEWERS' COMMENTS:

Reviewer #1 (Remarks to the Author):

The authors have adequately addressed my comments. They added a second N-cycle model to demonstrate the robustness of their original conclusions, and this is an excellent addition to the paper. I recommend the paper be published.

Reviewer #1's assessment of the response to Reviewer #2's comments:

I believe the authors have adequately addressed reviewer 2 comments and I recommend publications. The following is a point by point assessment of the authors' response to the issues raised.

Paragraph 1

no significant response required.

Paragraph 2 - "The entire result of this model ..."

First, it is phosphate, not nitrate that is the long term limiting nutrient because N-fixation and denitrification can change the amount of nitrate in the system. Therefore it is possible that phosphate cycling and NCP may not change (i.e. phosphate export and remineralisation stays constant) irrespective of how nitrate levels change in the ocean

Second, the link between global Nitrate and global NCP is more likely a negative correlation than a positive one. The modelling work by a co-author (Buchanan 2019 in review in Nature Comms) shows this, and in this study, it reflects the importance of phosphate and iron in controlling the ability of N-fixers to exploit conditions when the water has no nitrate. The relationship is dependent on the N-cycle model which controls the distribution of N fixation and denitrification.

Third, as stated in the authors' response, the relationship between global nitrate concentration and global NCP is complicated because of regional variability. At steady state, N fixation must balance denitrification, and this links the distribution of oxygen in the ocean with the spatial patterns of N fixation and denitrification.

Paragraph 3- "In this model run..."

The model response behaves as expected. Under sufficient phosphate and no nitrate the N-fixing phytoplankton dominate because they can utilise the N_2 dissolved in the water as their nitrate source. The nitrate and phosphate concentrations set the N fixation preference.

Par 4- "The authors explain ..."

I'm satisfied the authors' addressed this issue by considering a second N-cycle model, which showed a consistent behaviour to the original model. Behaviours is robust across two very different N-cycle formulations.

Par 5 "What would happen if NCP ..."

The authors' response adequately addressed the issue raised. I'm not surprised by the simulate behaviour. To support their response, the Buchanan paper that explored a suite of N-cycles also showed a negative correlation between global nitrate and global NCP.